# Spoiler Detection as Semantic Text Matching

**Ryan Tran*, Canwen Xu*, Julian McAuley**
University of California, San Diego
{rhtran,cxu,jmcauley}@ucsd.edu

## Abstract

Engaging with discussion of TV shows online often requires individuals to refrain from consuming show-related content for extended periods to avoid spoilers. While existing research on spoiler detection shows promising results in safeguarding viewers from general spoilers, it fails to address the issue of users abstaining from show-related content during their watch. This is primarily because the definition of a spoiler varies depending on the viewer's progress in the show, and conventional spoiler detection methods lack the granularity to capture this complexity. To tackle this challenge, we propose the task of *spoiler matching*, which involves assigning an episode number to a spoiler given a specific TV show. We frame this task as semantic text matching and introduce a dataset comprised of comments and episode summaries to evaluate model performance. Given the length of each example, our dataset can also serve as a benchmark for long-range language models.[1] [2]

## 1 Introduction

Many online social platforms (e.g., Reddit, Discord) provide opportunities for fans to discuss a particular TV show and share their thoughts about details in episodes. However, engaging in such discussions comes with the risk of spoilers, which may lead to unsatisfactory viewing experiences (Johnson and Rosenbaum, 2015). As a result, many viewers avoid these communities altogether until they have caught up with the latest episode of the show.

As an attempt to resolve this problem, some platforms (such as Reddit) have built-in functionality allowing users to tag their content as containing

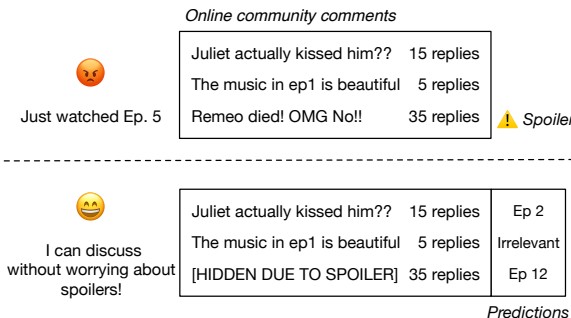

Figure 1: The task of spoiler matching. The model matches the comments that discuss a plot to the corresponding episode summary.

a spoiler. However, for long shows with a large number of episodes, this proves unsatisfactory: A user could be halfway through a show but will still be afraid to click on spoiler-tagged content for fear that it might contain spoilers for events later in the show when in reality, it might pertain to events that the viewer has already seen and with which they can engage. Again, some websites allow users to tag spoilers with more granularity, but it is far from guaranteed that users will be both accurate and consistent in tagging their content. This highlights the need for automatic *spoiler matching*. Unlike traditional spoiler detection (Boyd-Graber et al., 2013; Wan et al., 2019; Chang et al., 2021; Wróblewska et al., 2021), which determines whether a comment is a spoiler, spoiler matching aims to match a given spoiler to an episode number. A spoiler matching model working hand-in-hand with a spoiler detection model could provide much more fine-grained protection from spoilers.

In this work, we consider the setting where the show is known, and we would like to determine the episode to which a comment is referring, as shown in Figure 1. Specifically, we pose the problem as a semantic text matching (Cer et al., 2018) task between comments and episode summaries. We obtain a high-quality evaluation dataset by manually

---

*Equal contribution.

[1]Code and model weights are publicly available at https://github.com/bobotran/spoiler-matching

[2]The data is available at https://huggingface.co/datasets/bobotran/spoiler-matching

| Summary |
| --- |
| "...After thinking back to Yor's training, Anya uses her "killer move" and throws the ball at Bill. However, the ball hits the ground and bounces toward Bill, who throws the ball right back and hits her. Bill and his team were excited, thinking he was going to get a Stella Star. However, Henry informs them that they do not give out Stella Stars for a simple P.E. game..." |

| Comments |
| --- |
| **Relevant - Same episode:** "Anya: 'Finisher strike: Star Catch Arrow!!' Ball: 'nah, i don't really feel like it' "
**Relevant - Different Episode:** "The dog finally has a name. Borf!"
**Irrelevant:** "This episode was fun. Just joy from start to end."
**Irrelevant:** "Haven't been this hyped over a dodgeball game since Hunter x Hunter." |

Table 1: Example comments with an example summary from the show *Spy × Family*. The first two comments are *relevant*; The first one corresponds to the same episode as the summary while the second does not, so they are examples of a positive and negative example (respectively) during matching training. The third and fourth comments are *irrelevant* and are filtered out of the dataset during the auto-labeling step.

labeling the posts with the corresponding episode. To obtain a larger training set, we use prompt learning to efficiently fine-tune an auto-labeler model to automatically label another 200k examples, in addition to 5.9k manually labeled examples. The task is to match the comments to the correct episodes, with pairs of {episode summary, comment} as the input. As the number of episodes in a show is often limited, this task form is practically feasible. Also, as the median length of the concatenated summary and comment is 1,538 tokens, it can be a good dataset for benchmarking long-range Transformer models (Beltagy et al., 2020; Zaheer et al., 2020; Xiong et al., 2021).

## 2 Spoiler Matching Dataset

**Data Collection** We crawl 522,991 comments across 13 TV shows from discussion threads on Reddit. We also crawl 496 episode summaries from their respective episode pages on the website Fandom.[3] Reddit threads have a hierarchical discussion format where users reply to each other's posts, and each reply increases the indentation level. Top-level comments are comments that are not made in reply to any other comment. Threads focused around discussion of a particular episode are hand-picked by the human annotators, and the top-level comments are scraped. We take only top-level comments to minimize collecting comments that are incomplete thoughts continued from another part of a conversation. We then remove links and markup elements and apply other cleaning preprocessing before proceeding.

**Manual Labeling** After data cleaning, we group the comments by show name and episode number

| Method | F1 | Recall | Precision | AUC |
| --- | --- | --- | --- | --- |
| Fine-Tuning | 79.75 | 78.45 | 81.09 | 90.23 |
| LM-BFF (2021) | 81.09 | 84.55 | 77.90 | 90.92 |
| LM-BFF Ensemble | 81.62 | 86.18 | 77.51 | 91.56 |

Table 2: Test set performance of the relevant/irrelevant auto-labeler. For each model, the threshold that resulted in the highest F1 score on the validation set is chosen and used to compute the test set F1, recall, and precision.

based on the discussion thread from which they are scraped, but we do not yet consider them labeled at this step. This is because we find that about half of the comments scraped this way are irrelevant. We take this time to note the subtle difference in labels at this step compared to traditional spoiler detection.

While traditional spoiler detection classifies text as spoiler or non-spoiler, we at this step look to separate the irrelevant comments we have scraped from the relevant ones. We define a relevant comment as one that describes events from the episode (discussion thread) from which it is scraped. Examples of irrelevant comments are discussions about music, acting quality, personal feelings about the episode, etc. Table 1 shows some examples. While the first irrelevant comment is straightforward, the second is a more nuanced example: The episode in question is about dodgeball, but the comment does not discuss any events that occurred in the episode, so it is not considered relevant. Finally, we manually label 11,032 comments, reserved for the validation and test set.

**Automatic Labeling** To obtain a larger-scale training set, we split the 11,032 manually labeled examples into a small training, validation, and test set by the ratio of 7:2:1, to train an auto-

[3]https://fandom.com

labeler. To maximize data efficiency with this small dataset, we perform prompt-based fine-tuning to train RoBERTa (Liu et al., 2019). Specifically, we use LM-BFF (Gao et al., 2021) to fine-tune the model to predict if the comment is relevant.

We evaluate LM-BFF under two settings. For the first, we perform prompt-based fine-tuning on a set of hand-crafted templates, then select the best one according to validation set AUC. For the second, we use the automatic template search (*auto-T*) to generate and rank 40 templates based on validation AUC. Under both settings, we do not perform the *auto-L* search and instead fix the label words as "relevant" and "irrelevant". From this ranking, the top 20 templates are selected, and the logits from the corresponding models are averaged. Table 2 shows the evaluation of various auto-labelers. We find LM-BFF with template ensembling to be effective for our use case. Critically, a higher recall score means more irrelevant comments caught and filtered, resulting in a cleaner downstream training set for matching.

It is important to note that at this step, summaries are not concatenated with comments before being fed to the auto-labeler; the auto-labeler predicts relevant/irrelevant based on the words in the comment alone. This is because the job of the auto-labeler is to filter out generic irrelevant comments; unlike matching, episode-specific context is not required to perform this task.

Using our LM-BFF Ensemble auto-labeler, the 511,959 unlabeled comments are auto-labeled, separating 204,475 relevant comments from 307,484 irrelevant ones. Using our test set, we estimate that about 12% of the relevant comments are actually irrelevant. As we will demonstrate later, this number is low enough that the auto-labeled comments still serve as an effective training set for fine-tuning a spoiler matching model.

**Dataset Construction**    To recap, we have 496 episode summaries, 511,959 auto-labeled comments and 11,032 hand-labeled comments. Among the auto-labeled comments, we have 204,475 relevant comments and among the hand-labeled we have 5,892. Relevant comments are converted to the spoiler matching dataset format by assigning them the episode number of the discussion thread from which they were scraped. To test the ability of the matching models to generalize to unseen shows, the test set is constructed such that it contains 3,105 hand-labeled comments from 4 shows that are nei-

| #Examples | #Tokens | | |
|---|---|---|---|
| | 25% | 50% | 75% |
| Summary | 496 | 1136.5 | 1537.5 | 2035.25 |
| Comment | 210,367 | 18 | 28 | 51 |

Table 3: Statistics of the Spoiler Matching Dataset.

| Method | Human Labels Only | | w/ Auto-labels | |
|---|---|---|---|---|
| | dev | test | dev | test |
| BM25 | 38.90 | 36.80 | 40.67 | 36.80 |
| RoBERTa | 54.63 | 31.03 | 50.40 | 35.19 |
| Nyströmformer | 56.74 | 33.83 | 49.23 | 40.16 |
| BigBird | 62.75 | 33.28 | 54.25 | 46.22 |
| MaxP-RoBERTa | 63.42 | 40.44 | 58.63 | 54.09 |
| Longformer | 65.57 | **42.71** | 64.40 | **61.09** |

Table 4:  MRR of the baseline methods on Spoiler Matching Dataset. Note that the development sets of the two settings are different thus the numbers on the development sets are not comparable.

ther in the validation set nor the training set. The remaining 2,787 hand-labeled comments are used for validation. The statistics of the resulting dataset are shown in Table 3.

## 3   Experiments

**Task Formulation**    Our task is to match the comment to a certain episode in the show. Since there are only a limited number of episodes in a show, we iterate through all episode summaries of the show and concatenate each episode summary with the comment with a special token (e.g., [SEP]) inserted. After inference for each summary-comment pair, we rank the episodes by the predicted matching scores. We use Mean Reciprocal Rank (MRR, Craswell, 2009) as the metric for evaluation.

**Baselines**    We select 6 baselines, including BM25 as a lexical matching baseline, RoBERTa (Liu et al., 2019) as a pretrained language model baseline, and three long-range pretrained Transformers: BigBird (Zaheer et al., 2020), LongFormer (Beltagy et al., 2020), and Nystromformer (Xiong et al., 2021). We truncate the summary if the input exceeds the maximum tokens allowed for each model. For RoBERTa, we also experiment with the MaxP passage aggregation strategy (Dai and Callan, 2019). All models are base size. We implement the training and evaluation pipeline with Hugging Face Transformers (Wolf et al., 2020). The models are fine-tuned with the AdamW optimizer,

| Rank | Correct | Prediction | Matching Score | Comment |
|---|---|---|---|---|
| 23 | 21 | 1 | 0.1031 | Great santa still alive |
| 1 | 2 | 1 | 0.5822 | Kars the ultimate lifeform is released from his stone inprisonment !!!! |
| 1 | 11 | 13 | 0.8801 | I really really want Anya to have a dog she can communicate with. If she has a pupner with the ability to predict the future........ |

Table 5: Longformer predictions on three validation set examples. For each given comment, the first column represents the rank of the correct episode; the second is the correct episode number; the third is the episode with the highest matching score; the fourth is this highest matching score, and the fifth is the text of the comment. The score is the positive-class confidence after softmax.

a batch size of 32 and learning rate of 2e-5.

**Settings** We experiment with two settings to verify the effectiveness of the auto-labeling. In the human labels-only setting, we re-split the validation set to a small training set of 2,229 comments and a validation set of 558 comments. The second setting uses 204,475 automatically labeled training examples and 2,787 manually labeled examples for validation. The two settings share the same test set, which contains 3,105 manually labeled examples from 4 unseen shows.

**Experimental Results** Experimental results are shown in Table 4. In the human labels only setting, BM25 outperforms language models except MaxP-RoBERTa and Longformer. This finding is consistent with prior studies on text retrieval (Thakur et al., 2021) that BM25 can be a strong baseline when the training examples are insufficient. Compared with the human labels only setting, auto-labeling successfully improves model performance, verifying the effectiveness of our auto-labeling pipeline. Also, we observe that RoBERTa suffers from a short context length and underperforms BM25 under both settings. This is especially evident considering the large improvement resulting from the adoption of the passage aggregation strategy. Indeed, MaxP-RoBERTa is competitive with the long-range models. Among the long-range models, Longformer (Beltagy et al., 2020) achieves the best performance and outperforms the other models by a large margin. We hope our dataset can also serve as a benchmark for long-range pretrained language models, as the task requires interactions with long sentences.

## 4 Case Study

In this section, we analyze comments from two shows in the validation set, *Dr. Stone* and *Spy ×*

*Family* to understand behaviors of the models.[4]

Table 5 lists several challenging comments along with Longformer's output. The first example refers to a scene where Santa very briefly flies across the sky in the background. It is treated as unimportant, fantastical garnish on an event from the episode: None of the characters acknowledge it, so it is understood to have not actually occurred. Thus, the episode 21 summary does not mention it at all. This represents a class of comments that references relevant but obscure events, which only a recent viewer of the episode might remember. Interestingly, the score is relatively low, suggesting that the model understands that it is outputting a low-quality prediction.

The second comment describes an event from the show but references a character from an entirely different show; it is drawing a comparison between two characters, one of them in the show, based on physical likeness. The reference is fairly well-known within the community, but without additional external information, it would be difficult for the model to understand this comment beyond just context clues.

The third comment is challenging because it concerns predictions. Comments on ongoing shows often contain predictions, and if they happen to be correct, will likely match better lexically to the future episode when the events occur than the current episode when they are foreshadowed/predicted. This is not necessarily a bad thing for the end user, but it poses a challenge for training and evaluating our models. For this example, it is visually hinted in episode 11 that the dog has the ability to see the future but not confirmed until episode 13, so the summary for episode 11 does not mention this ability explicitly but the summary for episode 13 does, posing a possible explanation for the model's behavior.

---

[4]This section contains spoilers from these two shows.

Taken together, these examples give a glimpse into the challenges posed by spoiler matching. The hope is that this analysis motivates new lines of work into the study.

## 5 Conclusion

In this work, we define a new task of spoiler matching that formulates spoiler detection as a semantic text matching task and construct a large scale dataset with 223k comments and 496 episode summaries by mixing human and automatic labeling. We benchmark the performance of BM25 and four language models on the proposed dataset.

## Limitations

One limitation of our work is that due to resource restrictions, we only benchmark four language models on our dataset, leaving many other long-range language models untested. Also, our dataset only covers 13 shows and the comments annotated by human annotators are relatively limited.

## Acknowledgement

The authors would like to thank Ching Tan and Minh Nguyen for their help with annotating the validation set.

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
