# OpenReview forum: "Spoiler Detection as Semantic Text Matching"
_EMNLP/2023/Conference — EMNLP 2023 Main_

### Official Review · Reviewer_qhzf · 2023-07-31

**Typos Grammar Style And Presentation Improvements:** I have not found any.
**Soundness:** 4

**Excitement:**

4: Strong: This paper deepens the understanding of some phenomenon or lowers the barriers to an existing research direction.

**Missing References:**

I have not found any.

**Paper Topic And Main Contributions:**

The paper defines a novel NLP task of spoiler matching, which links a particular spoiler to a related specific episode from a TV show; moreover, it presents a dataset for this task (text episodes' summaries and related user comments).

**Questions For The Authors:**

I like the paper very much! The descriptions are bright and clear to read. I have no questions.

**Reasons To Accept:**

- novel and interesting NLP task and dataset
- a smart procedure for preparing the dataset with a very interesting description
- also, the dataset has another usage for long-range language models, which was tested and described in the paper
- very interesting and convincing result analysis with examples

**Reasons To Reject:**

I have not found any :-)


**Reproducibility:**

4: Could mostly reproduce the results, but there may be some variation because of sample variance or minor variations in their interpretation of the protocol or method.

**Reviewer Confidence:**

4: Quite sure. I tried to check the important points carefully. It's unlikely, though conceivable, that I missed something that should affect my ratings.

---

> ### Author Rebuttal · Authors · 2023-08-29
>
> We would like to thank you for your endorsements!

---

### Official Review · Reviewer_W1pv · 2023-08-05

**Soundness:** 3

**Excitement:**

4: Strong: This paper deepens the understanding of some phenomenon or lowers the barriers to an existing research direction.

**Justification For Ethical Concerns:**

N.A.

**Paper Topic And Main Contributions:**

The paper introduces the problem of identifying spoilers in discussion platforms like Reddit. The problem posed here is not one of detecting a spoiler but matching a spoiler to the episodes. Their key contributions are:

1: Creating a spoiler matching dataset.
2: Create a two step algorithm for identifying irrelevant comments first and then identifying spoilers from among them.
3: Benchmark performance of BM25 and four other models.

**Reasons To Accept:**

1. The paper is well explained.
2. Case study identifies good challenges in the task.

Since this is a dataset paper, I am recommending an accept.

**Reasons To Reject:**

Due to the following concerns I am proposing a rejection for this paper.

1: There is no need to make this a two step solution - first identifying irrelevant / relevant and then identifying episode numbers. One could either solve it as a multiclass problem with "irrelevant" or "other" being one class if we were doing this in traditional ML ways. in the LLM world, same prompt can address both these cases (identifying whether comment is relevant and matching comment to episode).

2: Their claim that auto-labeler identifying irrelevant comments does not need episode specific context is invalid. In fact, an argument can be made that their performance (F1 80.09) can be improved by adding this information.

3: Poor performance of auto-labeler - Given the scope of the problem, I believe 77% precision and 81% recall are too low specially since this is being used as auto-labeler. Techniques like ensemble, prompt engineering, active learning all could have been applied to improve this.

4: I am concerned about the novelty of the approach itself - in essence this is a multiclass semantic matching problem easily solvable through few shot prompt engineering.

Update:
the authors have addressed the comments in their rebuttal, I encourage them to add these explanations in the paper.

**Reproducibility:**

5: Could easily reproduce the results.

**Reviewer Confidence:**

5: Positive that my evaluation is correct. I read the paper very carefully and I am very familiar with related work.

**Typos Grammar Style And Presentation Improvements:**

I suggest a re-write of the abstract. Personally it sounded a bit convoluted and hard to follow.

---

> ### Author Rebuttal · Authors · 2023-08-29
>
> Thanks for your review. We would like to clarify our design choices and hope this can address your concerns.
>
> 1. **Re. Ranking setting**: We agree that adding “irrelevant” as a label instead of filtering them out is feasible. However, the decision to filter out the irrelevant posts is to align our dataset with a classic text matching/ranking setting where each text example is matched to a passage (i.e., episode summary in our paper).
> 2. **Re. Episode context for auto-labeler**: Thanks for your suggestion. We did try to add episode context when training the auto-labeler and it has a negative impact on performance. (1) One possible reason is as shown in Table 4, adding episode context will make the sequence too long and introduce noise. (2) Adding the episode context could lead to overfitting the seen episodes.
> The results of an auto-labeler trained with episode context are shown in the table below. As recall is the metric of interest for filtering out irrelevant examples, adding episode context does not improve the quality of data.
>
> | Method                 | F1    | Recall | Precision | AUC   |
> |------------------------|-------|--------|-----------|-------|
> | LM-BFF                 | 81.09 | 84.55  | 77.90     | 90.92 |
> | LM-BFF+Episode Context | 80.82 | 80.08  | 81.57     | 91.84 |
>
> 3. **Re. Poor performance of the auto-labeler**: (1) Our auto-labeler can indeed improve performance of the matching model by a large margin, as shown in Table 4, which verifies its effectiveness. (2) We will release our raw data and annotations clearly marked as “auto label” or “manual label” so researchers can develop their own auto-labeler. Meanwhile, our test set stays the same for consistent evaluation, as all data are manually labeled. \
> 	We’ve also run an experiment to ensemble the auto-labeler models. The results of the ensemble of 20 models are shown in the table below. Ensemble can slightly improve the auto-labeler’s performance.
>
> | Method            | F1    | Recall | Precision | AUC   |
> |-------------------|-------|--------|-----------|-------|
> | LM-BFF            | 81.09 | 84.55  | 77.90     | 90.92 |
> | LM-BFF (ensemble) | 81.62 | 86.18  | 77.51     | 91.56 |
>
> 4. **Re. Novelty concerns**: We would like to highlight the contribution of our dataset, which is suitable for long-form text matching. We would also like to point out that in production, using generative LLMs (e.g., ChatGPT or GPT-4) for text matching is financially infeasible, as every comment-summary pair requires one call of the API. That being said, as all test examples are manually labeled, our dataset could be a benchmark to evaluate few-shot learning for LLM research.

---

### Official Review · Reviewer_SNpa · 2023-08-06

**Soundness:** 3

**Excitement:**

3: Ambivalent: It has merits (e.g., it reports state-of-the-art results, the idea is nice), but there are key weaknesses (e.g., it describes incremental work), and it can significantly benefit from another round of revision. However, I won't object to accepting it if my co-reviewers champion it.

**Paper Topic And Main Contributions:**

This paper proposes to formalize the spoiler detection task into a semantic text matching problem, and demonstrated the feasibility of applying off-the-shelf LM-based text matching models on this tasks. The major contribution is the problem formalization and the corresponding dataset construction. A dataset of 5,892 hand-labeled and 271,881 auto-labeled relevant comments to 496 episodes of 13 tv shows is collected, and the authors indicate this dataset, together will the code and model will be made public.

**Reasons To Accept:**

- Spoiler often leads to unpleasant experiences during shows-related discussions on social media, spoiler detection studied in this paper is an interesting yet complex problem yet to be properly addressed.
- A dataset with a mix of hand-labels and auto-labels is constructed, and will benefit the related studies in the community.
- It sounds reasonable to formalize spoiler detection tasks into semantic text matching problems.

**Reasons To Reject:**

- The test set contains 3105 hand-labeled relevant comments from 4 unseen shows, yet only 5,892 hand-labeled relevant comments available for the 13 shows in total. Does this mean the data set is highly biased or imbalanced at least? If yes, this would make the evaluation results less convincible, wouldn't it?
- Summary truncation could lose some key facts making the matching results incomplete for RoBERTa, but some passage aggregation strategy (e.g., rrf) could easily mitigate this in table 4.
- Although the problem has been formalized into a text matching problem, precision and recall still sound more meaningful than MRR as evaluation metrics, I'd be more interested in seeing those numbers in Table 4.

**Reproducibility:**

3: Could reproduce the results with some difficulty. The settings of parameters are underspecified or subjectively determined; the training/evaluation data are not widely available.

**Reviewer Confidence:**

3: Pretty sure, but there's a chance I missed something. Although I have a good feel for this area in general, I did not carefully check the paper's details, e.g., the math, experimental design, or novelty.

---

> ### Author Rebuttal · Authors · 2023-08-29
>
> Thank you for your insightful comments.
>
> 1. **Re. Unbalanced dataset splits**: We intended to manually label all data for the test set to ensure the quality of data for evaluating the models. For shows in the training set, we only label a small portion of them and use auto-labeler to complete the rest. As few-shot and zero-shot methods are more and more common, we believe that our decision to allocate more effort to test set annotation aligns with this line of research.
> 2. **Re. Summarization for RoBERTa**: Thanks for your suggestion. Truncation is a simple strategy and default behavior for libraries including Hugging Face Transformer. We agree that a more complex strategy can improve the performance over truncation. We used MaxP [Dai et al., 2019] to aggregate the scores over passages from the episode summary. This aggregation strategy improves the performance by a large margin. We believe such methods can also be competitive to long text Transformers and our benchmark can be a testbed for these methods.
>
> | Method       | Human Labels Only (dev) | Human Labels Only (test) | w/ Auto Labels (dev) | w/ Auto Labels (test) |
> |--------------|-------------------------|--------------------------|----------------------|-----------------------|
> | RoBERTa      | 54.63                   | 31.03                    | 45.14                | 34.99                 |
> | MaxP-RoBERTa | **63.42**               | **40.44**                | **57.71**            | **53.69**             |
>
> 3. **Re. Precision &  recall**: Thanks for your suggestion. We’d like to note that since there is only one relevant item, MRR is equal to mean average precision (MAP). We will update our table to include separate precision and recall numbers.
>
> ----
> **[Dai et al., 2019]** Deeper Text Understanding for IR with Contextual Neural Language Modeling

---

### Meta-Review · Area_Chair_8JUq · 2023-09-18

**Recommendation:** 5

**Metareview:**

This paper describes the creation of a spoiler-to-episode matching dataset. Different from previous datasets which focused on spoiler detection as a binary task, this dataset offers the possibility to match datasets with specific episodes. This makes a novel and valuable contribution to the task of spoiler detection.

I would suggest fleshing out the paper with extra details as suggested by reviewers, adding more evaluation metrics (precision and recall) and more details on how the data was sampled.

I would also suggest to make the contribution of a novel dataset (not just a new task) more explicit in the introduction.

Does the current paper title really convey the new task / dataset that it proposes?

---

### Decision · Program_Chairs · 2023-10-07

**Decision:**

Accept-Main

**Comment:**

This paper describes the creation of a spoiler-to-episode matching dataset. Different from previous datasets which focused on spoiler detection as a binary task, this dataset offers the possibility to match datasets with specific episodes. This makes a novel and valuable contribution to the task of spoiler detection.

I would suggest fleshing out the paper with extra details as suggested by reviewers, adding more evaluation metrics (precision and recall) and more details on how the data was sampled.

I would also suggest to make the contribution of a novel dataset (not just a new task) more explicit in the introduction.

Does the current paper title really convey the new task / dataset that it proposes?